# PROTECTING MEMBERSHIP PRIVACY THROUGH ADAPTIVE LOGIT SCALING

## ABSTRACT

Machine learning (ML) models are susceptible to membership inference attacks (MIAs), where adversaries attempt to determine whether a specific data point is part of the model's training data. Recent studies suggest that MIAs often exploit the model's overconfidence in predicting training samples, albeit using various proxy indicators. To mitigate this vulnerability, we introduce Adaptive Logit Scaling (**ALS**) loss, a simple yet effective modification to the standard Cross-Entropy loss. ALS adaptively constrains the norm of the output logits for each sample during training by decoupling and dynamically scaling overly large logits based on their magnitudes. The proposed approach reduces the models' overconfidence and ensures that they produce less distinguishable output metrics between member and non-member data. Extensive evaluations across four benchmark datasets show that ALS consistently achieves strong membership privacy while maintaining high model accuracy. Further comparisons with nine state-of-the-art defenses against eight MIAs demonstrate that ALS effectively optimizes both sides of the privacy-utility trade-off, offering an effective and practical defense against MIAs.

## 1 INTRODUCTION

Machine learning (ML) models have demonstrated remarkable success in a wide range of applications. However, their widespread deployment raises significant privacy concerns, particularly in domains involving sensitive data, such as healthcare (Erickson et al., 2017; Ran et al., 2023) and facial recognition (Parkhi et al., 2015). One critical threat is membership inference attacks (MIAs) (Shokri et al., 2017), where adversaries attempt to determine whether a specific data point was part of the model's training dataset.

A major enabler of MIAs is model overconfidence (Shokri et al., 2017; Salem et al., 2018; Chen & Pattabiraman, 2024)—the tendency of deep learning models to assign excessively high confidence scores to training samples compared to unseen data. This overconfidence behavior creates a clear statistical distinction between training (member) and non-training (non-member) data, allowing adversaries to exploit output statistics such as confidence scores (Song & Mittal, 2021; Yeom et al., 2018), entropy (Shokri et al., 2017; Yeom et al., 2018), or loss values (Yeom et al., 2018) to infer data membership. While numerous defense mechanisms have been proposed to mitigate membership privacy risks (Nasr et al., 2018; Jia et al., 2019; Shejwalkar & Houmansadr, 2021; Tang et al., 2022; Abadi et al., 2016; Song & Mittal, 2021; Chen et al., 2022), many come at the cost of reduced model utility, require auxiliary data that may not always be accessible, or impose additional computational overhead, limiting their practicality.

This paper proposes Adaptive Logit Scaling (ALS) loss, a novel loss function designed to mitigate model overconfidence by adaptively constraining the output logit norms during training. Unlike the standard Cross-Entropy loss, which allows logits to grow arbitrarily large, ALS decouples the impact of the logit norm and adaptively regulates the output logit magnitude for each training sample during optimization, thereby preventing the model from assigning overly confident predictions and reducing the distinguishability between member and non-member samples while preserving high model accuracy. Our contributions can be summarized as follows: (1) We introduce Adaptive Logit Scaling (ALS) loss, a simple yet effective modification to Cross-Entropy loss that adaptively constrains logit magnitudes during training to mitigate overconfidence, thereby reducing membership inference risks. (2) Extensive experiments on four benchmark datasets show that models trained with

ALS achieve state-of-the-art membership privacy protection while preserving high model accuracy. (3) We further compare ALS against eight existing defense methods, demonstrating that it effectively balances the trade-off between privacy and utility (i.e., achieving strong membership inference privacy without significantly compromising the models' prediction performance), outperforming the baselines without requiring extra data and incurring additional computational overhead.

## 2 RELATED WORK

**Membership Inference Attacks**. MIAs (Shokri et al., 2017) seek to identify whether a specific data sample $x$ was included in the training dataset of a machine learning model. Given access to a target model $h$, an adversary constructs an inference classifier $\mathcal{A}$ to predict membership $\mathcal{A}(x; h) \in \{0, 1\}$, where $\mathcal{A}(x; h) = 1$ if $x$ is a training sample (member), and $\mathcal{A}(x; h) = 0$ otherwise (non-member). Existing MIAs exploit statistical differences in a model's output behavior by leveraging various output proxies, and they can be broadly categorized into score-based attacks and label-only attacks.

In score-based attacks, adversaries exploit the target model's output, such as loss values (Yeom et al., 2018), confidence scores (Song & Mittal, 2021; Yeom et al., 2018), entropy scores (Shokri et al., 2017; Yeom et al., 2018), modified-entropy scores (Song & Mittal, 2021). These output values serve as indicators for distinguishing between member and non-member samples, either through fixed thresholds or learned classifiers (e.g., neural networks (NN)-based attack (Nasr et al., 2019)). The Likelihood Ratio Attack (LiRA) (Carlini et al., 2022) represents the state-of-the-art score-based attacks. LiRA trains $N$ shadow models, half with the target samples and half without them. It then approximates their output distributions using Gaussian distributions and applies a likelihood-ratio test to infer membership.

In label-only attacks, the adversary has access only to the predicted class label. These attacks rely on the observation that training members tend to be more robust to perturbations or transformations than non-members. For example, the prediction-correctness attack (Yeom et al., 2018) labels any misclassified data point as a non-member. The boundary attack (Choquette-Choo et al., 2021; Li & Zhang, 2021) utilizes adversarial examples such as CW2 (Carlini & Wagner, 2017) to measure the distance of samples to the target model's decision boundary, inferring membership based on a threshold $\tau$. Augmentation attacks (Choquette-Choo et al., 2021) members' resilience to data augmentations, such as translations, and infer the membership of the query input $x$ based on the target model's classification consistency across augmented versions.

**Defenses Against MIAs**. Multiple defenses have been developed to mitigate MIAs. They can be grouped into several categories. Adversarial methods, such as Adversarial Regularization (Nasr et al., 2018) and MemGuard (Jia et al., 2019) aim to reduce privacy risks by employing adversarial training or perturbations to obscure membership signals in the model outputs. Knowledge distillation approaches such as DMP (Shejwalkar & Houmansadr, 2021) and SELENA (Tang et al., 2022) improve privacy by training teacher models on private datasets and transferring their knowledge to student models, thus hiding private information in the teacher models. Differential privacy techniques, such as DPSGD (Abadi et al., 2016), train models with a formal privacy guarantee by injecting noise into gradients during optimization. Regularization-based techniques aim to reduce overfitting, a primary enabler of MIAs (Truex et al., 2019). Methods like label smoothing(Guo et al., 2017) and early stopping(Caruana et al., 2000; Song & Mittal, 2021) improve generalization and decrease the model's tendency to memorize training data.

Several novel loss functions have been proposed to directly target overfitting and overconfidence in the context of MIA defense. For instance, RelaxLoss (Chen et al., 2022) applies gradient ascent to samples prone to overfitting during training to discourage memorization. However, this approach slows down model convergence and reduces model utility. HAMP (Chen & Pattabiraman, 2024) trains models with high-entropy soft labels and incorporates entropy regularization into the loss function to encourage output uncertainty. While effective in reducing privacy leakage, HAMP introduces additional computational overhead at inference time. It requires an extra forward pass on random noise and ranks its output relative to the original prediction to obtain the final result.

**Broader Impact of Overconfidence**. Beyond privacy leakage, overconfidence in model predictions is closely linked to poor calibration (Guo et al., 2017), which undermines the reliability of predicted probabilities. Poorly calibrated models are more vulnerable to adversarial examples, less effective at

out-of-distribution (OOD) detection (Wei et al., 2022), and more prone to causing factual hallucinations in large language models (Tian et al., 2023). Building on prior efforts in improving confidence calibration (Guo et al., 2017; Müller et al., 2019; Wei et al., 2022), our approach focuses on reducing overconfidence to improve membership privacy and maintain strong model utility without incurring additional data or inference overhead.

## 3 METHODOLOGY

### 3.1 PROBLEM STATEMENT

Overfitting in machine learning models has been demonstrated as a primary factor of membership privacy leakage (Salem et al., 2018; Shokri et al., 2017). MIAs exploit overfitting using various strategies to infer whether a specific data point was used during training. (Chen & Pattabiraman, 2024) offers a unifying perspective, arguing that the root cause underlying MIAs is overconfidence, where training samples are assigned disproportionately higher confidence scores than unseen data. A theoretical explanation for why neural networks trained with the standard softmax Cross-Entropy (CE) loss often produce overconfident outputs is provided in (Wei et al., 2022). The key insight is that overconfidence stems from the large magnitude of the neural network's output logits. This inflated output scaling amplifies confidence scores to produce overconfidence in predictions, particularly for training examples.

Let $\boldsymbol{f}$ denote the network output $f(\boldsymbol{x}; \theta)$, where $\boldsymbol{x}$ represents the input and $\theta$ corresponds to the model parameters. This output, also referred to as the logit vector $\boldsymbol{f}$ or pre-softmax output, can be decomposed into two components:

$$\boldsymbol{f} = \|\boldsymbol{f}\| \cdot \hat{\boldsymbol{f}}, \tag{1}$$

where $\|\boldsymbol{f}\| = \sqrt{\boldsymbol{f}_1^2 + \boldsymbol{f}_2^2 + \cdots + \boldsymbol{f}_k^2}$ is the Euclidean norm (*magnitude*) of the logit vector $\|\boldsymbol{f}\|$, and $\hat{\boldsymbol{f}}$ is the corresponding unit vector indicating its *direction*. The standard Cross-Entropy loss can then be reformulated as

$$\mathcal{L}_{\text{CE}}(f(\boldsymbol{x}; \theta), y) = -\log p(y|\boldsymbol{x}) = -\log \frac{e^{\|\boldsymbol{f}\| \cdot \hat{f}_y}}{\sum_{i=1}^{k} e^{\|\boldsymbol{f}\| \cdot \hat{f}_i}}.$$

This expression reveals that the training loss depends on both the magnitude $\|\boldsymbol{f}\|$ and the direction $\hat{\boldsymbol{f}}$ of the logit vector. When the model correctly classifies an input $y = \arg\max_i(f_i)$, further increasing $\|\boldsymbol{f}\|$ results in an increase in $p(y|\boldsymbol{x})$, the predicted probability confidence for the correct class, and subsequently minimizes the loss. This process leads the model to amplify the logit magnitudes for training examples, which results in highly confident predictions. As shown in **Figure 1a**, this behavior causes models trained with Cross-Entropy loss to assign near-maximal confidence (close to 1.0) to training samples, making them easily distinguishable from non-members.

Building on the above analysis, Wei et al. (2022) proposed a normalization-based strategy called LogitNorm to constrain the magnitude of the logit vector during training. Specifically, they enforce a fixed $L_2$ norm by rescaling the logits before applying the softmax, which leads to the following modified loss:

$$\mathcal{L}(f(\boldsymbol{x}; \theta), y) = -\log \frac{e^{f_y/\|\boldsymbol{f}\|}}{\sum_{i=1}^{k} e^{f_i/\|\boldsymbol{f}\|}}, \tag{2}$$

This formulation can be generalized by introducing a temperature parameter $\tau$ that controls the scaling of logits (with $\tau = 1$ in Equation 2 as a special case). Reducing the logit magnitude has been shown to improve robustness to out-of-distribution (OOD) inputs by mitigating overconfidence.

However, **in the context of privacy protection**, this fixed-norm approach has a key limitation: it applies the same scaling factor $\tau$ to all samples, neglecting the fact that certain training points are more prone to memorization and overfitting than others. As a result, a one-size-fits-all normalization strategy may not adequately mitigate privacy risks for the most vulnerable data points.

### 3.2 ADAPTIVE LOGIT SCALING

To address this limitation, we propose *Adaptive Logit Scaling* (ALS) loss, which introduces an adaptive penalty that dynamically adjusts the scaling strength based on the magnitude of the logits.

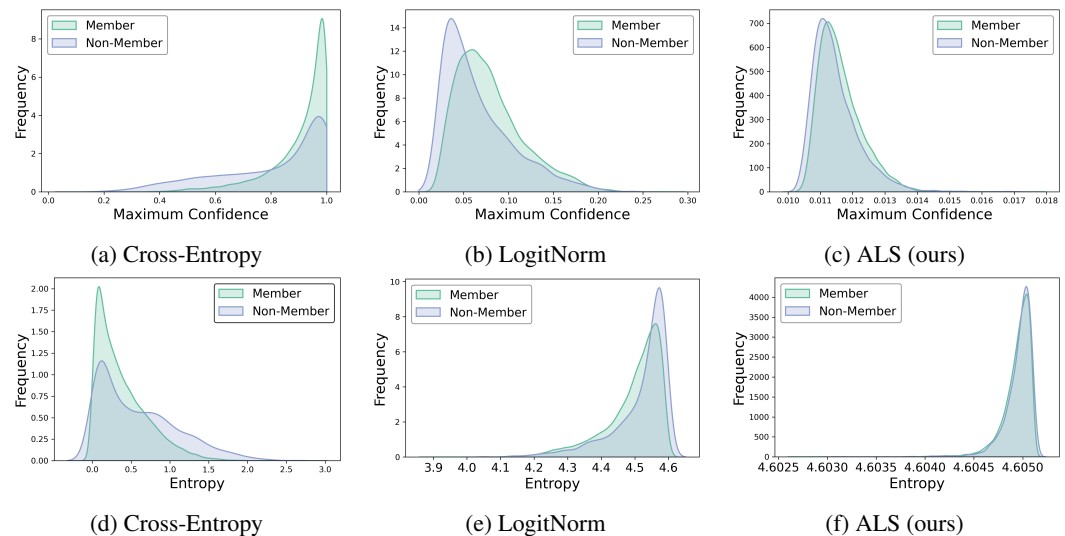

Figure 1: Output distribution of member and non-member data on the Purchase100 dataset. Figures (a), (b), and (c) show the distributions of the maximum softmax confidence score for models trained with Cross-Entropy loss, Logitnorm, and our proposed ALS loss, respectively. Figures (d), (e), and (f) present the corresponding output entropy distributions for models trained with Cross-Entropy loss, Logitnorm, and our proposed ALS loss, respectively.

The core intuition is that samples with larger logit norms are more likely to overfit. By adaptively modulating the scaling, ALS suppresses overconfident predictions on memorized data without compromising overall model utility. In addition, we incorporate an entropy regularization term that encourages high-entropy outputs, preventing overconfidence. Formally, the objective function of ALS is defined as:

$$\mathcal{L}_{\text{ALS}}(f(\boldsymbol{x};\theta),y) = -\log \frac{e^{f_y/(\alpha\|\boldsymbol{f}\|)}}{\sum_{i=1}^{k} e^{f_i/(\alpha\|\boldsymbol{f}\|)}} - \lambda\mathbb{H}(\sigma(\boldsymbol{f}/\|\boldsymbol{f}\|)), \tag{3}$$

where $\sigma(\cdot)$ is the **softmax** function, $\mathbb{H}(\sigma(\boldsymbol{f}/\|\boldsymbol{f}\|))$ denotes the entropy of the output, and $\lambda$ is a hyperparameter that controls the entropy regularization. The temperature $\alpha$ is an adaptive function of the logit norm, defined as $\alpha = 1 + \log(1 + \frac{\|\boldsymbol{f}\|}{s})$, with $s$ controlling the sensitivity of the scaling, i.e., how strongly $\alpha$ responds to changes in $\|\boldsymbol{f}\|$. Smaller $s$ values cause $\alpha$ to increase more rapidly, whereas larger $s$ dampen this effect. This adaptive design ensures that samples with larger logit norms yield larger $\alpha$, which in turn yields more uniform predicted probabilities and mitigates overconfidence.

We illustrate the effects of training models with ALS and compare them with those trained under Cross-Entropy and LogitNorm losses in **Figure 1**. In contrast to models trained with Cross-Entropy loss (**Figure 1a**) and LogitNorm (**Figure 1b**), the confidence scores of training members under ALS (**Figure 1c**) are significantly reduced, bringing them closer to those of non-members. This result demonstrates that ALS can effectively mitigate overconfidence. Similarly, **Figure 1f** shows that the entropy distributions of predictions for members and non-members are more closely aligned under ALS compared to those of Cross-Entropy loss (**Figure 1d**) and LogitNorm (**Figure 1e**). This result indicates that ALS can effectively reduce the distinguishability between the member and non-member data. Importantly, while both LogitNorm and ALS can drive entropy toward its theoretical maximum (e.g., $\log(100) = 4.06$ for Purchase100 with 100 classes), ALS achieves this without degrading predictive utility, as we will demonstrate in Section 4.

By incorporating adaptive logit scaling into ALS, the magnitude of the output vector of each data is strictly restricted to a value no greater than 1, as the minimum value of the scaling temperature $\tau$ is 1. Therefore, optimization under ALS (Eq. 3) focuses on adjusting the direction of the logit vector $f$ instead of increasing its magnitude. As a result, the model tends to produce more conservative predictions, particularly for inputs that are far away from the model's decision boundary (i.e., overfitted training samples). To formally characterize this behavior and exploit ALS against overconfidence, we derive a theoretical lower bound for the loss function defined in Eq.(3).

**Proposition 3.1** (Lower Bound of Loss). *For any input $\boldsymbol{x}$ and any positive number $\tau \in \mathbb{R}^+$, the per-sample loss defined in Eq. (2) has a lower bound: $\mathcal{L}_{ALS} \geq \log k^{-\lambda} \left(1 + (k-1)e^{-2}\right)$, where $k$ is the number of classes and $\lambda$ is the hyperparameter of the entropy regularization.*

The proof is detailed in Appendix A. This proposition demonstrates that there is a lower bound for adaptive logit scaling. This lower bound is determined by the hyperparameter $\lambda$ and the number of classes $k$. For example, when $k = 10$ and $\lambda = 0.1$, the lower bound is approximately 0.5769.

**Inference-time scaling.** During inference, we additionally apply logit scaling to the model's outputs to further reduce the separability between member and non-member samples, making their output distributions more indistinguishable. This operation involves only a single scaling computation and thus introduces virtually no additional inference overhead.

## 4 EXPERIMENTS

### 4.1 SETUPS

**Datasets**. We consider four benchmark datasets: Purchase100 Shokri et al. (2017), Texas100 Shokri et al. (2017), CIFAR10 and CIFAR100 Krizhevsky et al. (2009). Following prior work Shejwalkar & Houmansadr (2021); Chen & Pattabiraman (2024), the training splits are as follows: 20,000 samples for Purchase100, 15,000 for Texas100, and 25,000 each for CIFAR10 and CIFAR100. Additional dataset details are provided in Appendix B.

**Models**. For Purchase100 and Texas100, we use a fully connected (FC) network. For CIFAR10 and CIFAR100, we use DenseNet100 Huang et al. (2017).

**Attacks**. We evaluate ALS against eight MIAs. For NN-based attacks, we employ the black-box NSH attack Nasr et al. (2019), which leverages model loss, logit, and ground-truth labels to train the attack inference model. Additionally, we consider the loss-based attack Yeom et al. (2018), as well as confidence-, entropy-, and modified-entropy-based attacks described in Song & Mittal (2021). For Likelihood Ratio Attack (LiRA) Carlini et al. (2022), we train 128 shadow models, with 64 IN and 64 OUT models tailored to the defense. Following Chen & Pattabiraman (2024), we also implement boundary- and augmentation-based attacks in Choquette-Choo et al. (2021). Specifically, we employ the CW2 attack Carlini & Wagner (2017) to generate adversarial samples and determine the distance threshold for distinguishing members from non-members. For augmentation-based attacks, we apply translation-based augmentation.

**Defense baselines**. We compare ALS with nine MIA defenses: AdvReg Nasr et al. (2018), Mem-Guard Jia et al. (2019), DMP Shejwalkar & Houmansadr (2021), SELENA Tang et al. (2022), Label Smoothing (LS) Szegedy et al. (2016), DPSGD Abadi et al. (2016), HAMP Chen & Pattabiraman (2024), RelaxLoss Chen et al. (2022), and LogitNorm (Wei et al., 2022). While LogitNorm was originally proposed for OOD detection, we also adapt it as a baseline for comparison. Unless otherwise stated, all baseline defenses are implemented as described in Chen & Pattabiraman (2024), more details can be found in Appendix C.

**Evaluation Metrics**. An effective defense against MIAs should protect both members and non-members. Following established evaluation standards Carlini et al. (2022), we consider two metrics: (1) attack True Positive Rate evaluated at 0.1% False Positive Rate (Attack TPR @ 0.1% FPR) to measure the protection for members, and (2) attack True Negative Rate at 0.1% False Negative Rate (Attack TNR @ 0.1% FNR) to measure the protection for non-members.

### 4.2 RESULTS

**Models trained with ALS significantly reduce membership privacy leakage compared to those trained with Cross-Entropy**. The first two columns in **Figure 2** present the highest attack TPR@0.1% FPR and TNR@ 0.1% FNR across all four datasets. Compared to undefended models, those trained with ALS consistently achieve significantly lower MIA attack TPR@0.1% FPR and TNR@0.1% FNR. For example, the average highest attack TPR@ 0.1FPR% across the four datasets for undefended models is 8.18%, whereas ALS reduces it to merely 0.36%, representing a 95.6% reduction. Similarly, ALS reduces the attack TNR@ 0.1% FNR by 97.03%, from 14.16% in the undefended models to 0.42%. These results demonstrate the effectiveness of ALS in mitigating

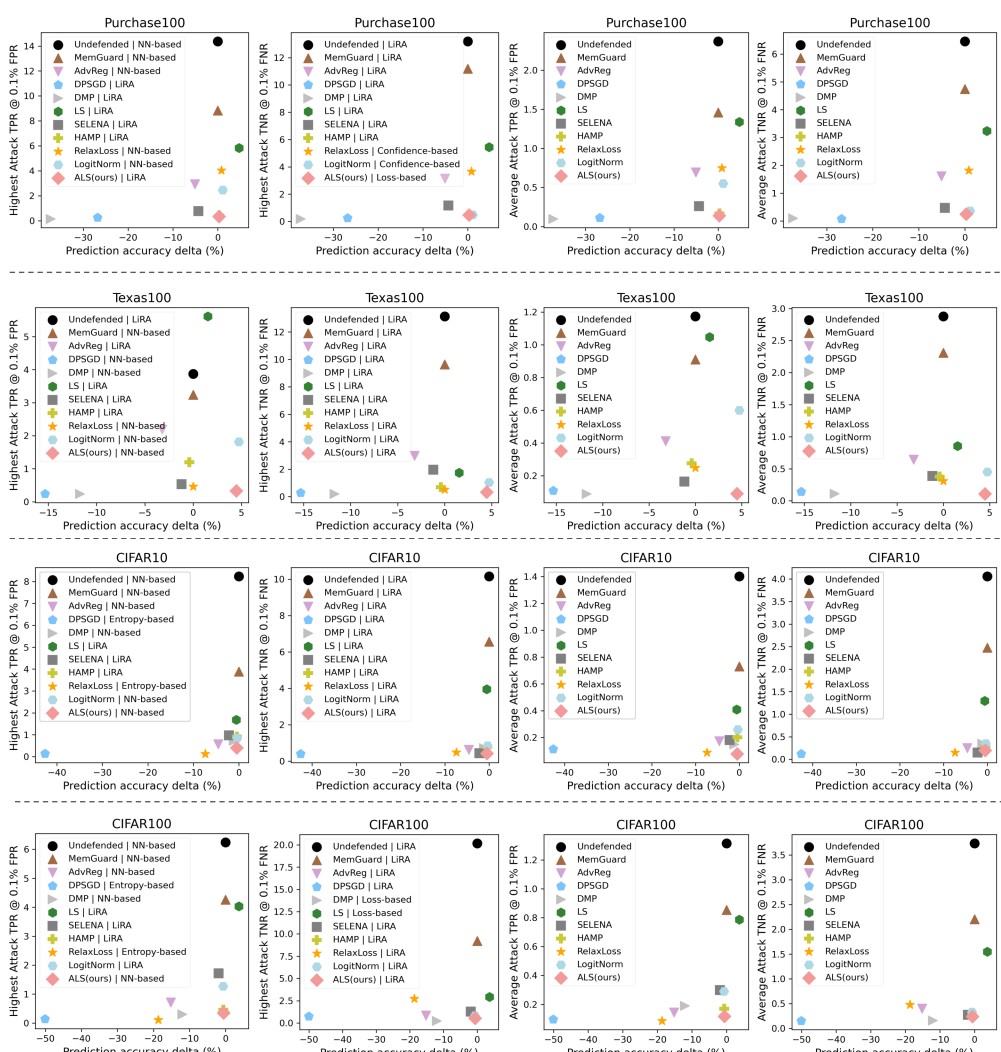

Figure 2: **Privacy-utility trade-off**. The first two columns present Defense|Attack with the highest TPR @ 0.1% FPR and the highest TNR@0.1% FNR in eight attacks for four datasets, respectively. Since we report the best-performing attack per defense, the selected attacks may differ. The third column shows the average Attack TPR@ 0.1% FPR across all eight attacks, while the last column presents the average Attack TNR@ 0.1% FNR. The x-axis is the prediction accuracy delta, which represents the change in accuracy relative to undefended models. An ideal defense should minimize accuracy loss while maintaining a low attack TPR and TNR, positioning toward the bottom right region of the figure. Overall, ALS consistently demonstrates strong privacy protection for both members and non-members while preserving high model accuracy, achieving a preferred privacy-utility trade-off compared to other defenses.

overconfidence, significantly limiting the adversary's ability to infer data membership in the target model. Detailed breakdowns of Figure 2 are provided in Table 5 and Table 6 in the Appendix.

Importantly, ALS achieves strong membership privacy while maintaining high model accuracy. Across all four datasets, ALS matches or exceeds the performance of the undefended model. As shown in **Table 1**, the test accuracy drops by only 0.48% and 0.59% on CIFAR10 and CIFAR100, respectively. On Purchase100 and Texas100, ALS even improves test accuracy by 0.25% and 4.47%.

**With comparable model utility to the state-of-the-art defense methods that preserve model utility, models trained with ALS consistently outperform them in membership privacy.** We next compare ALS with four state-of-the-art MIA defense baselines that preserve model utility with minimal accuracy drops or even improve test accuracy: MemGuard, LS, SELENA, and HAMP, as

Table 1: Test accuracy of each defense method across different datasets. ΔAcc. represents the accuracy difference compared to the undefended model. ALS preserves model utility on par with the undefended baseline, with minimal degradation and in some cases even improvement. Note that LS and LogitNorm achieve slightly higher utility but incur higher privacy leakage, as shown in Figure 2.

| Dataset | Metrics | Undefended | MemGuard | AdvReg | DPSGD | DMP | LS | SELENA | HAMP | RelaxLoss | LogitNorm | ALS (ours) |
|---|---|---|---|---|---|---|---|---|---|---|---|---|
| Purchase100 | Train Acc. | 99.36 | 99.36 | 93.97 | 61.06 | 49.57 | 99.54 | 85.19 | 91.12 | 97.96 | 99.45 | 97.06 |
|  | Test Acc. | 80.85 | 80.85 | 75.75 | 54.05 | 43.55 | 85.60 | 76.50 | 81.15 | 81.65 | 81.95 | 81.10 |
|  | Δ Acc. | 0.00 | 0.00 | -5.10 | -26.80 | -37.30 | +4.75 | -4.35 | +0.30 | +0.80 | +1.10 | +0.25 |
| Texas100 | Train Acc. | 76.79 | 76.79 | 62.76 | 43.08 | 46.92 | 75.52 | 58.58 | 68.56 | 66.64 | 78.43 | 77.76 |
|  | Test Acc. | 54.80 | 54.80 | 51.60 | 39.47 | 43.07 | 56.33 | 53.60 | 54.40 | 54.80 | 59.53 | 59.27 |
|  | Δ Acc. | 0.00 | 0.00 | -3.20 | -15.33 | -11.73 | +1.53 | -1.20 | -0.40 | 0.00 | +4.73 | +4.47 |
| CIFAR10 | Train Acc. | 98.72 | 98.72 | 86.73 | 44.24 | 91.08 | 97.63 | 86.86 | 95.88 | 82.72 | 98.58 | 98.19 |
|  | Test Acc. | 86.72 | 86.72 | 82.16 | 44.12 | 85.56 | 86.16 | 84.52 | 86.28 | 79.32 | 86.36 | 86.24 |
|  | Δ Acc. | 0.00 | 0.00 | -4.56 | -42.60 | -1.16 | -0.56 | -2.20 | -0.44 | -7.40 | -0.36 | -0.48 |
| CIFAR100 | Train Acc. | 86.21 | 86.21 | 55.78 | 9.46 | 53.37 | 88.80 | 62.15 | 68.44 | 45.47 | 94.12 | 89.08 |
|  | Test Acc. | 59.56 | 59.56 | 44.36 | 9.48 | 47.52 | 63.24 | 57.64 | 58.92 | 40.88 | 58.86 | 58.97 |
|  | ΔAcc. | 0.00 | 0.00 | -15.20 | -50.08 | -12.04 | +3.68 | -1.92 | -0.64 | -18.68 | -0.70 | -0.59 |
| - | Average Δ Acc. | 0.00 | 0.00 | -7.02 | -33.70 | -15.56 | +2.35 | -2.42 | -0.30 | -6.32 | +1.95 | +0.91 |

shown in Table 1 and Figure 2. Compared to MemGuard, ALS provides much stronger membership privacy by reducing the average attack TPR@ 0.1% FPR (averaged from the highest attack TPR@ 0.1% FPR across four datasets) by 14.1x (from 5.06% to 0.36%) and the average attack TNR@ 0.1% FNR by 21.8x (from 9.15% to 0.42%).

Regarding LS, the test accuracy is improved by 4.75% on Purchase100, 1.53% on Texas100, and 3.68% on CIFAR100, but slightly drops by 0.56% on CIFAR10. Despite its general accuracy improvements, the model trained with LS remains highly vulnerable to MIAs. The average attack TPR@ 0.1% FPR on the LS-trained models is 4.29%, 11.9x higher than that by ALS. Similarly, the average attack TNR@ 0.1% FNR on the LS-trained model is 8.4x higher than that of ALS.

SELENA experiences an accuracy drop of 4.35% on Purchase100, 1.2% on Texas100, 2.2% on CIFAR10, and 1.92% on CIFAR100. Compared to it, the average attack TPR@ 0.1% FPR and the average attack TNR@ 0.1% FNR of ALS are lower by 0.64% and 0.79%, respectively.

HAMP offers the strongest balance between privacy protection and model utility, with an average attack TPR@ 0.1% FPR of 0.75% and TNR@ 0.1% FNR of 0.59% across four datasets. It also improves test accuracy by 0.3% on Purchase100 while incurring minimal accuracy drops of 0.4%, 0.44%, and 0.64% on Texas100, CIFAR10, and CIFAR100, respectively. ALS performs on par with HAMP in balancing the privacy-utility trade-off but is significantly more efficient. HAMP relies on high-entropy soft label generation and requires dataset-specific tuning of the entropy threshold to maintain accuracy. Furthermore, HAMP introduces additional computational overhead due to its test-time defense mechanism, which involves generating random samples and performing an additional inference step to determine the final output.

**With comparable MIA privacy to the state-of-the-art defense methods that achieve the lowest attack TPR and TNR, models trained with ALS consistently achieve better model utility.** As illustrated in Table 1 and Figure 2, DPSGD and DMP consistently achieve low attack TPR@ 0.1% FPR and TNR@ 0.1% FNR across the four datasets. Specifically, DPSGD, trained with a privacy budget of $\epsilon = 4$, achieves the lowest attack TPR@ 0.1% FPR and TNR@ 0.1% FPR, with an average attack TPR of 0.2% and an average TNR of 0.42%, which are lower than those of ALS by 0.16% and the same as that of ALS, respectively. However, this strong privacy protection comes at the cost of significant degradation of model utility, with accuracy drops of 26.8% on Purchase100, 15.33% on Texas100, 42.6% on CIFAR10, and 50.08% on CIFAR100.

Similarly, DMP achieves an average attack TPR@ 0.1% FPR of 0.36% and an average TNR@ 0.1% FNR of 0.33%. While it only experiences a slight accuracy drop of 1.16% on CIFAR10, it suffers large accuracy reductions of 37.30% on Purchase100, 11.73% on Texas100, and 12.04% on CIFAR100. In contrast, ALS achieves near-optimal privacy with barely any accuracy drop across all four datasets, making it a more practical choice for balancing privacy and utility.

RelaxLoss achieves low attack TPRs @ 0.1% FPRs of 0.46%, 0.13%, and 0.11% on Texas100, CIFAR10, and CIFAR100, which are 0.1% higher, 0.28%, and 0.24% lower than those of ALS. Its attack TNRs@ 0.1% FNRs of 0.53%, 0.48%, and 2.74% on these datasets are 0.19%, 0.06%, and 2.31% higher than ALS's. However, RelaxLoss degrades model utility, reducing accuracy by 7.4% on CIFAR10 and 18.68% on CIFAR100—likely due to its repeated gradient ascent on overfitted training data. While it maintains utility on Purchase100, with 0.8% improvement compared to the

Undefended model, its privacy leakage is substantial, with attack TPR@ 0.1% FPR and TNR@ 0.1% FNR values 11.9x and 7.8x higher than those of ALS.

AdvReg achieves a low attack TPR@ 0.1% FPR of 0.57% and an attack TNR@ 0.1% FNR of 0.63% on CIFAR10, at the cost of a large accuracy drop of 4.56%. It also achieves a low attack TPR@ 0.1% FPR of 0.57% and an attack TNR@ 0.1% FNR of 0.71% and 0.85%, respectively on CIFAR100 while incurring a more significant accuracy degradation of -15.2%. Although its test accuracy only drops by 5.1% and 3.2% on Purchase100 and Texas100, respectively, Ad-vReg fails to provide sufficient privacy protection. Specifically, AdvReg achieves an average attack TPR@ 0.1% FPR of 2.56% and an average arrack TNR@ 0.1% FNR of 3.07% on these two datasets, which are 3.8× and 4.6× higher than those of ALS, respectively.

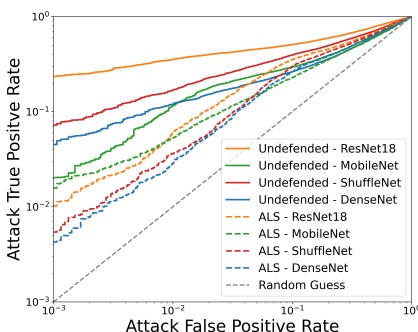

**ALS effectively enhances membership privacy across different architectures.** We next evaluate ALS on different model architectures, including ResNet18 He et al. (2016), MobileNet Howard (2017) and ShuffleNet Zhang et al. (2018), and DenseNet. We compare models trained with Cross-Entropy (Undefended) and ALS for each architecture under the same hyperparameter setting. **Figure 3** presents the attack ROC of different models. The results indicate that different models trained with Cross-entropy loss exhibit varying levels of vulnerability to MIAs, with attack TPR@ 0.1% FPR ranging from 2.03%~23.41%, yielding an average attack TPR@ 0.1% FPR of 9.30%. In contrast, models trained with ALS loss are able to mitigate

Figure 3: **Evaluation across different model architectures**. While models trained with different architectures exhibit varied degrees of MIA risks, ALS consistently reduces membership privacy leakage.

MIAs consistently, achieving attack TPR@ 0.1% FPR ranging 0.41%~1.57%, with an average attack TPR@ 0.1% FPR of 0.89%. On average, ALS reduces the attack TPR by 90.43% (from 9.30% to 0.89%). These results demonstrate that ALS provides strong protection against membership privacy leakage across different architectures.

## 5 DISCUSSION

**On the effectiveness against LiRA.** The results in Section 4 highlight ALS's effectiveness against different MIAs, including the state-of-the-art LiRA. Following the methodology of (Hayes et al., 2025), we examine the logit distribution of a specific sample from CIFAR-10 across the IN and OUT shadow models used in LiRA. Member logits correspond to shadow models trained on the sample, whereas non-member logits are obtained from shadow models that are not exposed to it during training. As shown in **Figure 4**, the two distributions exhibit substantial overlap, with only minor statistical differences. This further confirms the effectiveness of ALS, as ALS reduces the separability of member and non-member logit distributions from the shadow models, thereby effectively lowering the attack performance of LiRA.

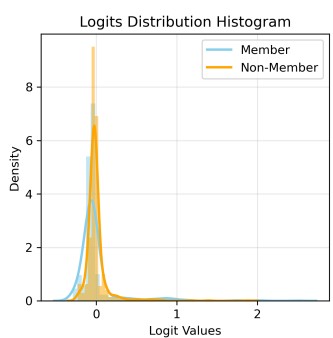

Figure 4: Logit distributions produced by shadow models for a specific CIFAR-10 sample.

**Effect of sensitivity** $s$. In **Figure 5**, we examine how the sensitivity $s$ in ALS affects both model utility and membership privacy on Purchase100 (results for Texas100 are provided in **Figure 6** in Appendix D). Specifically, we train models using the same hyperparameters while varying $s$ in the range $[0.1, 0.5, 1.0, 5.0, 10.0, 20.0, 30.0, 40.0]$. The reported attack TPR and TNR correspond to the attack that achieves the highest success rate in each setting.

As the scaling sensitivity parameter $s$ increases, test accuracy improves, but member privacy leakage also gets worse, reflected by higher attack TPR@ 0.1% FPR and TNR@ 0.1% FNR values. This trade-off arises because smaller $s$ values make the adaptive scaling more sensitive to the logit norm, leading to a stronger penalty that suppresses overconfident outputs. While this enhances privacy, it also makes optimization less effective. In the extreme case where $s \to \infty$, the adaptive scaling

Table 2: Performance of ALS with different entropy regularization $\lambda$ on Purchase100.

| $\lambda$ | Test Acc. | Attack TPR @0.1%FPR | Attack TNR @0.1%FNR |
|---|---|---|---|
| 0.5 | 77.86 | 0.11 | 0.18 |
| 0.1 | 79.41 | 0.21 | 0.28 |
| 0.05 | 79.45 | 0.27 | 0.23 |
| 0.01 | 80.30 | 0.31 | 0.29 |
| 0.005 | 80.10 | 0.34 | 0.25 |
| 0.001 | 81.20 | 0.44 | 0.32 |
| 0.0005 | 81.20 | 0.48 | 0.51 |
| 0.0001 | 81.35 | 0.63 | 0.59 |

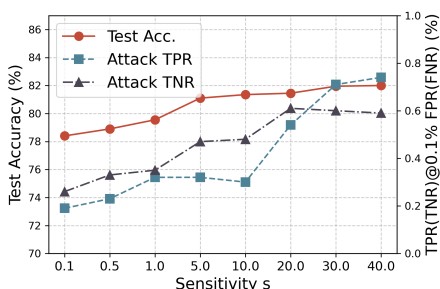

Figure 5: Model utility and privacy leakage vs. different sensitivity parameter $s$ on Purchase100.

factor $\alpha \to 1$, reducing the ALS formulation to Eq. 2, equivalent to the ALS (fixed) baseline in our ablation study. These results highlight the importance of carefully tuning the sensitivity parameter $s$ to balance privacy protection and model performance.

**Strength of entropy regularization** $\lambda$. To study the impact of entropy regularization. We vary its strength $\lambda$ in the range $[0.0001, 0.5]$. Increasing $\lambda$ encourages the model to produce higher-entropy (i.e., less confident) predictions, which helps mitigate overfitting and reduces susceptibility to MIAs. As shown in **Table 2**, stronger regularization with a larger $\lambda$ improves privacy protection, but at the cost of reduced test accuracy.

**Ablation Studies on Key Components**. As described in Section 3.2, ALS consists of three key components designed to enhance membership privacy: (1) entropy regularization, (2) inference-time scaling, and (3) adaptive scaling $\alpha$ during training. To understand the contribution of each component, we conduct ablation studies on the Purchase100 dataset. Specifically, we compare four configurations: (1) ALS-vanilla, which denotes ALS without entropy regularization and inference-time scaling; (2) ALS+reg, which denotes ALS with entropy regularization but without inference-time

Table 3: Ablation study under different settings on Purchase100.

| Setting | Attack TPR @0.1%FPR | Attack TPR @0.1%FPR |
|---|---|---|
| Undefended | 14.37 | 13.19 |
| ALS-vanilla | 2.13 | 1.26 |
| ALS+reg | 1.58 | 0.92 |
| ALS+scaling | 0.86 | 0.52 |
| ALS(fixed) | 0.76 | 0.58 |
| ALS(ours) | 0.32 | 0.47 |

scaling; (3) ALS+scaling, which denotes ALS with inference-time scaling but without entropy regularization; (4) ALS(fixed), which denotes ALS without adaptive scaling $\alpha$ but with both entropy regularization and inference-time scaling and (5) ALS, which is our complete setup.

**Table 3** reports the highest attack TPR@ 0.1%FPR and TNR@ 0.1%FNR across all attacks on Purchase100. Compared to the undefended model, training with ALS-vanilla reduces the TPR@ 0.1%FPR to 2.13% and the TNR@ 0.1%FNR to 1.26%, corresponding to reductions by 85.18% and 90.45%, respectively. Introducing entropy regularization further enhances privacy by promoting higher-entropy predictions, lowering the TPR@ 0.1%FPR to 1.58% and the TNR@0.1%FNR to 0.92%. In addition, inference-time scaling enhances privacy by flattening the output distribution, reducing the confidence gap between member and non-member examples. Importantly, this inference-time scaling does not affect the model's predicted class, as it applies only an input-dependent scaling factor to the logits. Finally, adaptive scaling during training provides additional privacy benefits over the fixed-scaling variant. Our full ALS implementation achieves the strongest privacy protection among all configurations by combining all three components effectively.

## 6 CONCLUSION

This paper introduced Adaptive Scaled Logits (ALS) loss, a simple yet effective modification to the standard Cross-Entropy loss that mitigates overconfidence and enhances membership privacy by constraining the norm of the output logits during training. ALS dynamically rescales logits based on their norms, decoupling magnitude from direction to suppress overconfident predictions and reduce the distinguishability between member and non-member data. Through comprehensive evaluations on four benchmark datasets and comparisons with nine state-of-the-art MIA defenses against eight MIAs, we demonstrated that ALS consistently achieves strong membership privacy protection while maintaining or even improving model accuracy.

## ETHICS STATEMENT

This work contributes to advancing membership privacy in machine learning by proposing an effective defense against membership inference attacks (MIAs) that balances privacy and utility. It offers practical benefits for safeguarding sensitive data in domains like healthcare and finance, where privacy breaches can be particularly harmful. However, the method does not eliminate all privacy leakage, and malicious adversaries may still exploit residual signals. We hope this research will inspire further work on privacy-preserving learning and promote the development of more robust defenses in real-world applications.

## REPRODUCIBILITY STATEMENT

We have provided detailed implementation descriptions within the paper to facilitate reproducibility. Additionally, we commit to releasing the source code and associated resources upon acceptance of this work.

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

## A  PROOF OF PROPOSITION 3.1

Let $\widetilde{\boldsymbol{f}} = \boldsymbol{f}/(\alpha\|\boldsymbol{f}\|)$, then we have $\|\widetilde{\boldsymbol{f}}\| = 1/\alpha$.

That is, $\sum_{i=1}^{k} \widetilde{\boldsymbol{f}}_i^2 = \|\widetilde{\boldsymbol{f}}\|^2 = 1/\alpha^2$.

Hence,

$$-1/\alpha \leq \widetilde{\boldsymbol{f}}_i \leq 1/\alpha, \forall\, i \in 1, \ldots, k.$$

Let $\sigma(\widetilde{\boldsymbol{f}}) = \frac{e^{\tilde{f}_y}}{\sum_{i=1}^{k} e^{\tilde{f}_i}}$, since $\alpha = 1 + \log(1 + \frac{\|\boldsymbol{f}\|}{s}) \geq 1$, then we have

$$
\begin{aligned}
\sigma(\widetilde{\boldsymbol{f}}) &\leq \frac{e^{1/\alpha}}{e^{1/\alpha} + (k-1)e^{-1/\alpha}} \\
&= \frac{1}{1 + (k-1)e^{-2/\alpha}} \\
&\leq \frac{1}{1 + (k-1)e^{-2}}
\end{aligned}
$$

Hence,

$$
\begin{aligned}
\mathcal{L}_{\text{ALS}} &= -\log(\sigma(\widetilde{\boldsymbol{f}})) - \lambda\mathbb{H}(\widetilde{\boldsymbol{f}}) \\
&\geq -\log\frac{1}{1 + (k-1)e^{-2}} - \lambda\log k \\
&= \log(1 + (k-1)e^{-2}) - \log k^{\lambda} \\
&= \log k^{-\lambda}(1 + (k-1)e^{-2})
\end{aligned}
$$

Thus Proposition 3.1 is proved. $\qquad\square$

## B  DATASETS

**Purchase100** Shokri et al. (2017) includes 197,324 customer shopping records, each with 600 binary features indicating whether a specific item was purchased. The classification task involves predicting a customer's shopping habits across 100 distinct classes.

**Texas100** Shokri et al. (2017) contains 67,330 hospital discharge records, each containing 6,170 binary features indicating whether the patient has a particular symptom or not. The objective is to predict the treatment category across 100 classes.

**CIFAR10** and **CIFAR100** Krizhevsky et al. (2009) are image classification datasets, each containing 60,000 images (32×32×3) spans across 10 and 100 classes, respectively.

Table 4: Training details for ALS.

| Dataset | $s$ | Epochs | Learning rate | Weight decay | Regulization $\lambda$ |
|---|---|---|---|---|---|
| Purchase100 | 5.0 | 40 | 5e-4 | 2e-4 | 0.005 |
| Texas100 | 0.5 | 10 | 5e-4 | 0.0 | 0.5 |
| CIFAR10 | 1.0 | 100 | 0.5 | 1e-4 | 0.001 |
| CIFAR100 | 1.0 | 100 | 0.5 | 1e-4 | 0.001 |

## C  IMPLEMENTATION DETAILS

**Training details of ALS**. We summarize the sensitivity $s$, learning rate, training epochs, and weight decay for Purchase100, Texas100, CIFAR10, and CIFAR100 in Table 4. All experiments are conducted on 2 NVIDIA GeForce RTX 4090.

**Baselines Implementation**. For all baselines, we follow the hyperparameter settings from the original paper wherever applicable, making only minimal adaptations to ensure consistency across evaluations. Specifically, for AdvReg, HAMP, RelaxLoss, and SELENA, we followed the hyperparameters in their original codebase for different datasets. For label smoothing, we follow (Kaya & Dumitras, 2021; Chen & Pattabiraman, 2024) to train LS with different smoothing intensities and select the model with the highest accuracy. Purchase100 is trained with a smoothing intensity of 0.03, Texas with 0.09, and CIFAR10 and CIFAR100 with 0.01. For DMP, we generated 200k synthetic tabular data with CT-GAN for both Purchase100 and Texas100 to train a reference model, and used DCGAN to generate 150k synthetic image data for CIFAR10 and CIFAR100 to train a reference model. For DPSGD, we used the implementation of DP-SGD using Pytorch Opacus (Kaya & Dumitras, 2021), and trained the models on all datasets with budget $\epsilon = 4$ following (Tang et al., 2022). For Memguard, we follow Chen & Pattabiraman, 2024 and use the provided models to generate adversarial perturbations during inference.

## D  ADDITIONAL RESULTS

**Table 5** and **Table 6** report the detailed performance of each defense method against individual attacks. The final column summarizes the average attack success rate across all attacks for each defense.

**ALS enhances training effectiveness and improves model utility. Table 7** presents the training and test accuracy of ResNet18, MobileNet, and ShuffleNet on CIFAR-10 using both Cross-Entropy (CE) loss and our proposed ALS loss. The results demonstrate that models trained with ALS consistently achieve higher test accuracy compared to those trained on CE. This improvement can be attributed to several factors. First, ALS introduces an implicit regularization by adaptively scaling the logits, which encourages the model to learn more generalizable representations and smoother decision boundaries, instead of relying on extreme activations for the classification. Second, in the case of ResNet18, ALS achieves lower training accuracy but higher test accuracy, indicating a reduction in overfitting. Conversely, for MobileNet and ShuffleNet, ALS improves both training and test accuracy, suggesting that the adaptive norm scaling of logits enforced by ALS stabilizes gradient computations during backpropagation, enabling more effective training.

**Comparing with temperature scaling**. Unlike ALS, which applies a data-dependent temperature during training to normalize each sample's logit norm, traditional temperature scaling uses a fixed global temperature at inference time to improve calibration. To evaluate whether global temperature scaling during training can help enhance privacy, we conduct experiments by tuning $\tau$ within $[2, 10]$ and selecting the value that minimizes attack TPR and TNR. **Table 8** in the Appendix D presents the results on all four datasets. We find that adding a global temperature scaling offers only marginal privacy gains over the undefended model and fails to effectively reduce overconfidence, as the output confidence scores for training data remain close to 1. In contrast, ALS dynamically adjusts the temperature per sample by normalizing the logit magnitude to 1, focusing optimization on the logit direction rather than its scale. This adaptive approach significantly reduces overconfidence and provides stronger privacy protection.

Table 5: Detailed attack TPR@0.1% FPR and TNR@0.1% FNR for each attack on Purchase100 and Texas100. **Bold** indicates the highest attack values for each defense.

| Dataset | Defense | Metric(%) | NN-based | Loss-based | Confidence-based | Entropy-based | M-entropy-based | Augmentation-based | Boundary-based | LiRA | Average |
|---|---|---|---|---|---|---|---|---|---|---|---|
| Purchase100 | Undefended | Attack TPR | **14.37** | 0.09 | 0.09 | 0.08 | 0.08 | N/A | 0.00 | 1.86 | 2.37 |
| | | Attack TNR | 0.20 | 10.24 | 10.24 | 1.70 | 9.64 | N/A | 0.00 | **13.19** | 6.46 |
| | MemGuard | Attack TPR | **8.84** | 0.05 | 0.05 | 0.05 | 0.05 | N/A | 0.00 | 1.18 | 1.46 |
| | | Attack TNR | 0.10 | 7.23 | 7.23 | 0.16 | 7.32 | N/A | 0.00 | **11.19** | 4.75 |
| | AdvReg | Attack TPR | **2.93** | 0.14 | 0.14 | 0.13 | 0.13 | N/A | 0.00 | 1.39 | 0.69 |
| | | Attack TNR | 0.16 | 2.77 | 2.77 | 0.58 | 1.86 | N/A | 0.00 | **3.14** | 1.61 |
| | DPSGD | Attack TPR | 0.04 | 0.11 | 0.11 | 0.14 | 0.12 | N/A | 0.00 | **0.26** | 0.11 |
| | | Attack TNR | 0.06 | 0.03 | 0.03 | 0.11 | 0.03 | N/A | 0.00 | **0.26** | 0.07 |
| | DMP | Attack TPR | 0.10 | 0.11 | 0.11 | 0.11 | 0.11 | N/A | 0.00 | **0.15** | 0.10 |
| | | Attack TNR | 0.08 | 0.10 | 0.10 | 0.10 | 0.09 | N/A | 0.00 | **0.19** | 0.09 |
| | LS | Attack TPR | 3.22 | 0.08 | 0.08 | 0.08 | 0.08 | N/A | 0.00 | **5.83** | 1.34 |
| | | Attack TNR | 0.14 | 5.07 | 5.07 | 1.69 | 5.26 | N/A | 0.00 | **5.44** | 3.24 |
| | SELENA | Attack TPR | 0.70 | 0.09 | 0.09 | 0.09 | 0.08 | N/A | 0.00 | **0.77** | 0.26 |
| | | Attack TNR | 0.06 | 0.67 | 0.67 | 0.15 | 0.56 | N/A | 0.00 | **1.17** | 0.47 |
| | HAMP | Attack TPR | 0.39 | 0.09 | 0.09 | 0.11 | 0.09 | N/A | 0.00 | **0.40** | 0.17 |
| | | Attack TNR | 0.13 | 0.35 | 0.35 | 0.08 | 0.33 | N/A | 0.00 | **0.44** | 0.24 |
| | RelaxLoss | Attack TPR | **4.04** | 0.18 | 0.18 | 0.21 | 0.18 | N/A | 0.00 | 0.44 | 0.75 |
| | | Attack TNR | 0.02 | **3.66** | **3.66** | 1.30 | 3.90 | N/A | 0.00 | 0.19 | 1.82 |
| | LogitNorm | Attack TPR | **2.47** | 0.24 | 0.24 | 0.19 | 0.24 | N/A | 0.00 | 0.45 | 0.55 |
| | | Attack TNR | 0.41 | 0.47 | **0.49** | 0.22 | **0.49** | N/A | 0.00 | 0.47 | 0.36 |
| | **ALS**(ours) | Attack TPR | 0.32 | 0.09 | 0.09 | 0.04 | 0.09 | N/A | 0.00 | **0.34** | **0.14** |
| | | Attack TNR | 0.24 | **0.47** | 0.46 | 0.06 | 0.46 | N/A | 0.00 | 0.04 | **0.25** |
| Texas100 | Undefended | Attack TPR | 3.67 | 0.17 | 0.17 | 0.16 | 0.17 | N/A | 0.00 | **3.87** | 1.17 |
| | | Attack TNR | 0.68 | 2.03 | 2.03 | 0.40 | 1.87 | N/A | 0.00 | **13.13** | 2.88 |
| | MemGaurd | Attack TPR | **3.24** | 0.13 | 0.13 | 0.08 | 0.14 | N/A | 0.00 | 2.65 | 0.91 |
| | | Attack TNR | 0.56 | 1.93 | 1.93 | 0.23 | 1.88 | N/A | 0.00 | **9.64** | 2.31 |
| | AdvReg | Attack TPR | 0.07 | 0.16 | 0.16 | 0.17 | 0.13 | N/A | 0.00 | **2.19** | 0.41 |
| | | Attack TNR | 0.25 | 0.39 | 0.39 | 0.10 | 0.37 | N/A | 0.00 | **2.99** | 0.64 |
| | DPSGD | Attack TPR | **0.24** | 0.10 | 0.10 | 0.10 | 0.10 | N/A | 0.00 | 0.13 | 0.11 |
| | | Attack TNR | 0.12 | 0.19 | 0.19 | 0.07 | 0.14 | N/A | 0.00 | **0.29** | 0.14 |
| | DMP | Attack TPR | **0.24** | 0.05 | 0.05 | 0.04 | 0.09 | N/A | 0.00 | 0.16 | 0.09 |
| | | Attack TNR | 0.04 | 0.13 | 0.13 | 0.13 | 0.15 | N/A | 0.00 | **0.21** | 0.11 |
| | LS | Attack TPR | 1.11 | 0.15 | 0.15 | 0.16 | 0.15 | N/A | 0.00 | **5.61** | 1.05 |
| | | Attack TNR | 0.62 | 1.03 | 1.03 | 0.59 | 0.97 | N/A | 0.00 | **1.75** | 0.86 |
| | SELENA | Attack TPR | 0.31 | 0.08 | 0.08 | 0.08 | 0.07 | N/A | 0.00 | **0.53** | 0.16 |
| | | Attack TNR | 0.13 | 0.16 | 0.16 | 0.19 | 0.10 | N/A | 0.00 | **1.97** | 0.39 |
| | HAMP | Attack TPR | 0.31 | 0.12 | 0.12 | 0.07 | 0.12 | N/A | 0.00 | **1.20** | 0.28 |
| | | Attack TNR | 0.07 | 0.59 | 0.59 | 0.11 | 0.59 | N/A | 0.00 | **0.70** | 0.38 |
| | RelaxLoss | Attack TPR | **0.46** | 0.21 | 0.21 | 0.21 | 0.21 | N/A | 0.00 | 0.44 | 0.25 |
| | | Attack TNR | 0.13 | 0.46 | 0.46 | 0.15 | 0.45 | N/A | 0.00 | **0.53** | 0.31 |
| | LogitNorm | Attack TPR | **1.82** | 0.18 | 0.18 | 0.18 | 0.18 | N/A | 0.00 | 1.69 | 0.60 |
| | | Attack TNR | 0.59 | 0.41 | 0.41 | 0.27 | 0.41 | N/A | 0.00 | **1.04** | 0.45 |
| | **ALS**(ours) | Attack TPR | **0.33** | 0.21 | 0.00 | 0.00 | 0.00 | N/A | 0.00 | 0.10 | **0.09** |
| | | Attack TNR | 0.03 | **0.34** | 0.00 | 0.01 | 0.26 | N/A | 0.00 | 0.09 | **0.11** |

Table 6: Detailed attack TPR@0.1% FPR and TNR@0.1% FNR for each attack on CIFAR10 and CIFAR100. **Bold** indicates the highest attack values for each defense.

| Dataset | Defense | Metric(%) | NN-based | Loss-based | Confidence-based | Entropy-based | M-entropy-based | Augmentation-based | Boundary-based | LiRA | Average |
|---|---|---|---|---|---|---|---|---|---|---|---|
| CIFAR10 | Undefended | Attack TPR | **8.23** | 0.00 | 0.00 | 0.10 | 0.00 | 0.02 | 0.10 | 2.76 | 1.40 |
| | | Attack TNR | 0.05 | 6.24 | 5.99 | 0.40 | 6.00 | 3.63 | 0.00 | **10.15** | 4.06 |
| | MemGaurd | Attack TPR | **3.89** | 0.08 | 0.08 | 0.07 | 0.09 | 0.02 | 0.10 | 1.52 | 0.73 |
| | | Attack TNR | 0.13 | 2.96 | 2.96 | 0.20 | 3.35 | 3.63 | 0.00 | **6.57** | 2.48 |
| | AdvReg | Attack TPR | **0.57** | 0.09 | 0.09 | 0.14 | 0.14 | 0.05 | 0.13 | 0.18 | 0.17 |
| | | Attack TNR | 0.04 | 0.40 | 0.30 | 0.16 | 0.29 | 0.18 | 0.00 | **0.63** | 0.25 |
| | DPSGD | Attack TPR | 0.10 | 0.13 | 0.13 | **0.14** | **0.14** | 0.08 | 0.07 | 0.12 | 0.11 |
| | | Attack TNR | 0.07 | 0.12 | 0.12 | 0.10 | 0.13 | 0.03 | 0.00 | **0.40** | 0.12 |
| | DMP | Attack TPR | **0.73** | 0.06 | 0.00 | 0.10 | 0.00 | 0.10 | 0.10 | 0.12 | 0.15 |
| | | Attack TNR | 0.05 | 0.37 | 0.67 | 0.12 | 0.68 | 0.20 | 0.00 | **0.72** | 0.35 |
| | LS | Attack TPR | 1.22 | 0.00 | 0.00 | 0.07 | 0.00 | 0.23 | 0.08 | **1.68** | 0.41 |
| | | Attack TNR | 0.09 | 1.85 | 1.85 | 0.30 | 2.07 | 0.27 | 0.00 | **3.95** | 1.30 |
| | SELENA | Attack TPR | 0.18 | 0.00 | 0.00 | 0.11 | 0.00 | 0.13 | 0.05 | **0.98** | 0.18 |
| | | Attack TNR | 0.07 | 0.16 | 0.11 | 0.19 | 0.06 | 0.16 | 0.00 | **0.43** | 0.15 |
| | HAMP | Attack TPR | 0.39 | 0.00 | 0.11 | 0.00 | 0.00 | 0.08 | 0.12 | **0.92** | 0.20 |
| | | Attack TNR | 0.17 | 0.17 | 0.26 | 0.00 | 0.51 | 0.38 | 0.00 | **0.77** | 0.28 |
| | RelaxLoss | Attack TPR | 0.00 | **0.13** | **0.13** | **0.13** | **0.13** | 0.05 | 0.02 | 0.12 | 0.09 |
| | | Attack TNR | 0.09 | 0.18 | 0.18 | 0.09 | 0.18 | 0.00 | 0.00 | **0.48** | 0.15 |
| | LogitNorm | Attack TPR | **0.85** | 0.12 | 0.12 | 0.12 | 0.12 | 0.00 | 0.07 | 0.57 | 0.26 |
| | | Attack TNR | 0.32 | 0.47 | 0.47 | 0.18 | 0.47 | 0.05 | 0.00 | **0.85** | 0.35 |
| | **ALS**(ours) | Attack TPR | **0.41** | 0.05 | 0.00 | 0.08 | 0.00 | 0.00 | 0.00 | 0.09 | **0.08** |
| | | Attack TNR | 0.12 | 0.23 | 0.16 | 0.25 | 0.35 | 0.03 | 0.02 | **0.42** | **0.20** |
| CIFAR100 | Undefended | Attack TPR | **6.24** | 0.09 | 0.13 | 0.14 | 0.15 | 0.07 | 0.12 | 3.57 | 1.31 |
| | | Attack TNR | 0.46 | 2.80 | 2.56 | 0.24 | 2.52 | 1.05 | 0.10 | **20.16** | 3.74 |
| | MemGaurd | Attack TPR | **4.26** | 0.15 | 0.15 | 0.12 | 0.10 | 0.07 | 0.12 | 1.86 | 0.85 |
| | | Attack TNR | 0.14 | 2.28 | 2.28 | 0.20 | 2.37 | 1.05 | 0.10 | **9.21** | 2.20 |
| | AdvReg | Attack TPR | **0.71** | 0.00 | 0.00 | 0.12 | 0.00 | 0.08 | 0.12 | 0.13 | 0.15 |
| | | Attack TNR | 0.31 | 0.60 | 0.57 | 0.08 | 0.65 | 0.12 | 0.00 | **0.85** | 0.40 |
| | DPSGD | Attack TPR | 0.10 | 0.10 | 0.10 | **0.14** | 0.09 | 0.07 | 0.08 | 0.09 | 0.10 |
| | | Attack TNR | 0.12 | 0.08 | 0.08 | 0.06 | 0.08 | 0.05 | 0.00 | **0.74** | 0.15 |
| | DMP | Attack TPR | **0.30** | 0.11 | 0.20 | 0.20 | 0.21 | 0.21 | 0.09 | 0.21 | 0.19 |
| | | Attack TNR | 0.11 | **0.24** | 0.20 | 0.17 | 0.23 | 0.12 | 0.00 | 0.21 | 0.16 |
| | LS | Attack TPR | 1.56 | 0.07 | 0.08 | 0.08 | 0.08 | 0.30 | 0.09 | **4.03** | 0.79 |
| | | Attack TNR | 0.28 | **2.92** | **2.92** | 0.29 | 2.71 | 0.37 | 0.00 | 2.88 | 1.55 |
| | SELENA | Attack TPR | 0.15 | 0.06 | 0.06 | 0.07 | 0.06 | 0.19 | 0.08 | **1.72** | 0.30 |
| | | Attack TNR | 0.15 | 0.21 | 0.19 | 0.11 | 0.20 | 0.06 | 0.00 | **1.28** | 0.28 |
| | HAMP | Attack TPR | 0.22 | 0.16 | 0.16 | 0.00 | 0.17 | 0.09 | 0.10 | **0.46** | 0.17 |
| | | Attack TNR | 0.08 | 0.41 | 0.41 | 0.07 | 0.41 | 0.24 | 0.00 | **0.47** | 0.26 |
| | RelaxLoss | Attack TPR | 0.09 | **0.11** | **0.11** | **0.12** | **0.11** | 0.09 | 0.00 | 0.07 | 0.09 |
| | | Attack TNR | 0.14 | 0.26 | 0.26 | 0.06 | 0.28 | 0.06 | 0.02 | **2.74** | 0.48 |
| | LogitNorm | Attack TPR | 0.48 | 0.12 | 0.12 | 0.12 | 0.12 | 0.11 | 0.00 | **1.27** | 0.29 |
| | | Attack TNR | 0.19 | 0.42 | 0.42 | 0.18 | 0.42 | 0.09 | 0.07 | **0.83** | 0.33 |
| | **ALS**(ours) | Attack TPR | **0.35** | 0.10 | 0.00 | 0.00 | 0.00 | 0.10 | 0.06 | 0.10 | **0.12** |
| | | Attack TNR | 0.08 | 0.33 | 0.00 | 0.21 | 0.29 | 0.11 | 0.00 | **0.43** | 0.24 |

Table 7: Train and Test accuracy (%) of different models on CIFAR10.

| Model | CE | | ALS | |
|---|---|---|---|---|
| | Train Acc. | Test Acc. | Train Acc. | Test Acc. |
| ResNet | 99.37 | 66.76 | 92.58 | 71.24 |
| MobileNet | 76.04 | 62.44 | 88.81 | 71.36 |
| ShuffleNet | 79.61 | 63.44 | 89.59 | 68.12 |

Table 8: Comparison with Temperature Scaling on Texas100, CIFAR10 and CIFAR100.

| Datasets | Method | Attack TPR @0.1%FPR | Attack TNR @0.1%FNR |
|---|---|---|---|
| Purchase100 | Undefended | 14.37 | 13.19 |
| | Temperature Scaling | 11.93 | 13.56 |
| | ALS(ours) | 0.32 | 0.47 |
| Texas100 | Undefended | 3.87 | 13.13 |
| | Temperature Scaling | 1.16 | 3.13 |
| | ALS(ours) | 0.33 | 0.34 |
| CIFAR10 | Undefended | 8.23 | 10.15 |
| | Temperature Scaling | 9.85 | 11.13 |
| | ALS(ours) | 0.41 | 0.42 |
| CIFAR100 | Undefended | 6.24 | 20.16 |
| | Temperature Scaling | 7.11 | 17.20 |
| | ALS(ours) | 0.35 | 0.43 |

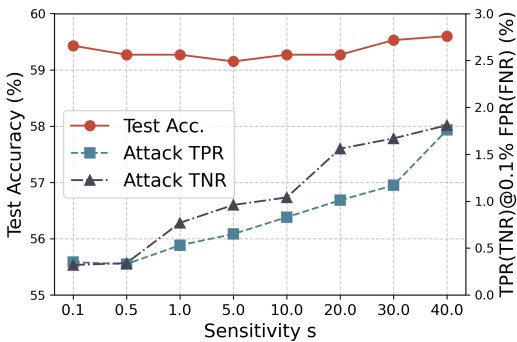

Figure 6: Model utility and privacy leakage v.s. different temperature parameter $\tau$ on Texas100.

