# OpenReview forum: "Protecting Membership Privacy through Adaptive Logit Scaling"
_ICLR.cc/2026/Conference — Submitted to ICLR 2026_

### Official Review · Reviewer_3gnU · 2025-10-27

**Soundness:** 2
**Presentation:** 3
**Contribution:** 2
**Rating:** 2
**Confidence:** 4

**Summary:**

This paper introduces an adaptive logit scaling loss, which scales large logits based on their magnitudes during training. Applying the loss to model training reduces the models’ overconfidence and produces less score distinction between member and non-member data, thereby mitigating the MIAs. Experiments show the performance of the method on various datasets.

**Strengths:**

1. The writing is clear and easy to follow. The paper clearly presents the proposed method through both mathematical formulations and explanations. The related work provides a comprehensive background that helps readers understand membership inference attacks. In addition, the presentation of the experimental results (including figures and tables) is clear and easy to follow.

2. Extensive experiments show that the performance of the proposed method is evaluated on various datasets across diverse attack and defence methods. The authors also conducted ablation studies on components of the method.

**Weaknesses:**

1. The paper demonstrates limited novelty. The authors incorporate LogitNorm, which was originally introduced to mitigate overconfidence and enhance out-of-distribution detection, into the training loss to defend against membership inference attacks. Although the authors add an entropy regularization term to LogitNorm, its function is similar to that of LogitNorm in mitigating overconfidence, as described in this paper.

2. The proposed method lacks a clear justification. At test time, the method requires applying a logit scaling (Is this temperature scaling?) to the model’s outputs. The author claims that the operation is to further reduce the separability between member and non-member data. However, with a sufficiently large temperature, the model can effectively mitigate overconfidence, producing predictions that are nearly uniform. If so, it is unclear why we need to independently add an ALS loss.

3. The paper does not provide a detailed experimental setup. For reproducibility, the paper should also provide the experimental parameters used for the baseline methods. The paper does not provide the AUC score, which is a key metric for evaluating membership inference attacks. In addition, the authors should plot privacy-utility curves to show the privacy-utility trade-offs, rather than reporting only a single point result as in Figure 2. Furthermore, the performance of ALS is comparable to that of baselines on CIFAR-10.

**Questions:**

1. L1, L2 regularization, and early stopping can also prevent overfitting. How do these methods perform in defending against MIAs compared to ALS?
2. How effective is ALS in defending against gradient-based attack methods?
3. How does ALS perform when non-members are OOD samples?
4. In ablation studies, how effective is using only inference-time scaling as a defence?

---

> ### Author Response · Authors · 2025-11-24
>
> Dear reviewer,
>
> We are thankful for your comments and would like to make the following clarifications.
>
> 1. **Regarding limited novelty.** We acknowledge and clearly state that our work is inspired by LogitNorm; however, LogitNorm was originally developed for OOD detection and has not been studied in the context of membership inference privacy. Its behavior and limitations under MIA were unknown. Our contribution is to systematically investigate its effectiveness for membership privacy, identify its shortcomings, and then develop an adaptive, data-dependent solution that specifically addresses these limitations. The proposed ALS is therefore not a simple adaptation but a purposefully designed mechanism tailored to reduce membership distinguishability, and our experiments demonstrate that it consistently improves privacy protection.
>
> 2. **Regarding temperature scaling.** In Appendix D, we explicitly compare ALS with traditional temperature scaling. Unlike ALS—which applies a **data-dependent, per-sample normalization during training**—standard temperature scaling uses a **fixed global temperature** applied only at inference for calibration purposes. Although very large temperatures can produce nearly uniform predictions, this does not alter the relative statistical gap between member and non-member outputs because the same global temperature is applied uniformly. As shown in Table 8, simply adding temperature scaling does **not** significantly improve privacy, whereas ALS yields substantial and consistent gains.
>
> 3. **Regarding detailed experimental setup and AUC score** . The full experimental setup including training configurations, hyperparameters, and attack specifications is provided in Appendix C. We will also release the source code upon acceptance for reproducibility. In addition, we report TPR@0.1%FPR and TNR@0.1%FNR following established standards [1,2].
>
> 4. **Regarding privacy-utility trade-off.** We would like to clarify that the privacy-utility trade-off we refer to in the paper highlights the challenges of achieving strong privacy protection without significantly compromising the model utility, which we have clearly mentioned in the paper. Figure 2 exactly presents this property for each defense method. In addition, as stated and justified clearly in the caption of Figure 2, **“since we report the best-performing attack per defense, the selected attacks may differ.”** This choice avoids underestimating any defense method by ensuring that each method is evaluated under its strongest adversary. Overall, ALS consistently provides both **strong privacy protection** for members and non-members and **high model utility**, achieving a more favorable privacy–utility trade-off compared with existing defenses-even when each is evaluated against its strongest attack. In addition, while the performance of ALS is comparable to that of baselines on CIFAR-10, ALS provided a **simple yet effective** solution without incurring additional computational overhead like HAMP does or complicating the training pipeline such as DMP and SELENA which adopt knowledge distillation.
>
> 5. **Regarding L1/L2 regularization and early stopping.** We would like to clarify that most defense baselines and our ALS models already incorporate standard techniques such as **weight decay (L2)** and **early stopping**, which are commonly used to reduce overfitting. Previous work[2] has also compared to early stopping  and shows that it only offer minor reductions in privacy leakage. Therefore, simple L1/L2 and early stopping are not able to achieve a favorable privacy–utility trade-off on their own.
>    **(Do not say we believe, cite some reference to say they already showed L1/L2 are weak in defending mia since the mia indicator is on the loss/confidence distribution difference)**
>
> 6. **Regarding gradient-based attack.**  We adopted IHA ((Inverse Hessian)  (https://github.com/iamgroot42/auditingmi/tree/main), which is one of the state-of-the-art white-box MIA and we follow their settings to train a 4-layer MLP on Purchase dataset using Cross-Entropy (Undefended) and the proposed ALS. We used the I-HVP approximation method and evaluated the attack using 500 member and 500 non-member samples. The results are summarized below:
>
>    |   Model    |  AUC  |
>    | :--------: | :---: |
>    | Undefended | 0.682 |
>    |    ALS     | 0.591 |
>
>    We report **AUC** as the evaluation metric, consistent with the original IHA codebase, since the attack outputs binary scores (0/1). These results indicate that ALS provides **improved privacy protection** over the undefended baseline, even against strong white-box attacks. However, the AUC of 0.591 still suggests susceptibility under this stronger threat model. We appreciate the reviewer’s suggestion and will consider incorporating white-box evaluations more comprehensively in future work, along with the design of more robust defenses.

---

> > ### Author Response · Authors · 2025-11-24
> >
> > 7. **Regarding OOD samples.** To further investigate how ALS performs under OOD samples. We conducted experiments with DenseNet100 trained on CIFAR10 and adopted SVHN as the OOD non-member data. The results are shown in the table below.
> >
> >    |      Model       | Highest TPR@0.1%FPR | Highest TNR@0.1%FNR |
> >    | :--------------: | :-----------------: | :-----------------: |
> >    | Undefended (ID)  |        14.37        |        13.19        |
> >    |     ALS (ID)     |        0.32         |        0.47         |
> >    | Undefended (OOD) |        17.29        |        15.16        |
> >    |    ALS (OOD)     |        0.40         |        0.58         |
> >
> >    The first two rows are the results reported in the main paper when non-member data are from the same distribution (i.e., CIFAR10). The last two rows show the results when the non-member samples come from an OOD dataset (i.e., SVHN). We observe that for the undefended model, privacy leakage becomes even more severe under the OOD setting, with both TPR and TNR increasing noticeably. In contrast, ALS still remain robust, exhibiting only a slight increase in leakage while still maintaining very low attack success rates. These results indicate that ALS can also preserve its privacy protection effectiveness even in challenging OOD scenarios.
> >
> >
> > 8. **Regarding more ablation study.** Following your suggestion, we additionally evaluate a setting where the model is trained with standard cross-entropy loss and **only inference-time scaling** is applied as a defense. The results are shown below:
> >
> > |            Model            | Highest TPR@0.1%FPR | Highest TNR@0.1%FNR |
> > | :-------------------------: | :-----------------: | :-----------------: |
> > |         Undefended          |        14.37        |        13.19        |
> > | Only inference-time scaling |        3.13         |        1.99         |
> > |             ALS             |        0.32         |        0.47         |
> >
> > As the results indicate, inference-time scaling alone provides limited privacy improvement over the undefended model. These results support the effectiveness of each component of ALS. We will also incorporate the new results in the camera-ready version.
> >
> >
> > [1] Carlini, Nicholas, et al. "Membership inference attacks from first principles." *2022 IEEE symposium on security and privacy (SP)*. IEEE, 2022.
> >
> > [2] Chen, Zitao, and Karthik Pattabiraman. "Overconfidence is a dangerous thing: Mitigating membership inference attacks by enforcing less confident prediction." *arXiv preprint arXiv:2307.01610* (2023).

---

### Official Review · Reviewer_zYSD · 2025-10-31

**Soundness:** 3
**Presentation:** 3
**Contribution:** 2
**Rating:** 4
**Confidence:** 4

**Summary:**

This paper studies defense mechanisms against membership inference attacks (MIAs) on classification models. In particular, it proposes to use Adaptive Logit Scaling (ALS), i.e., a novel loss function that dynamically adjusts the scaling of logits based on their magnitude and incorporates an entropy regularization term.
The key idea is that samples with larger logit norms are more likely to be overfitted and thus leak membership information. ALS penalizes these samples by rescaling their logits while encouraging higher output entropy. Experimental results show that this approach substantially reduces attack true positive rates (TPR) at low false positive rates (FPR, e.g., 0.1%) while maintaining competitive model utility.

**Strengths:**

- The presentation is generally clear, and the paper is overall well-structured.
- The proposed approach is lightweight and easy to implement.
- The experimental results are comprehensive.

**Weaknesses:**

- The general insight (penalizing overconfidence to improve privacy defense) and the proposed approach, which applies a form of confidence regularization, are not particularly novel. Moreover, the paper does not provide deeper insights that clearly distinguish it from the broader generalization literature.

- The experimental results are not very strong, as the improvements over existing methods  are marginal in many settings.

**Questions:**

- The role of Proposition 3.1 in the overall argument is unclear. Why is the lower bound of the loss relevant to privacy leakage or privacy defense? Even the standard cross-entropy loss is lower-bounded, yet it can still easily leak membership information.

- It appears that the paper still focuses primarily on standard (or “normal”) data samples, while overlooking literature [1] suggesting that such regularization-based defense approaches may be less effective for real privacy protection, particularly for “hard examples” which are often of greatest interest in the privacy context as they represent the worst-case scenarios.

- After reviewing some of the baseline literature, it seems confusing that the main shortcoming you attribute to RelaxLoss is its utility degradation, which contradicts the main findings of the original paper. Is there any specific reason why the same experimental settings or results could not be replicated in this work?

[1] Aerni, Michael, Jie Zhang, and Florian Tramèr. "Evaluations of machine learning privacy defenses are misleading." Proceedings of the 2024 on ACM SIGSAC Conference on Computer and Communications Security. 2024.

---

> ### Author Response · Authors · 2025-11-24
>
> Dear reviewer,
>
> We appreciate your constructive comments. We would like to make the following clarifications.
>
> 1. **Regarding the novelty of general insight.**  We acknowledge that penalizing overconfidence has been explored in prior work. However, our contribution is that we proposed an adaptive logits scaling loss formulation tailored specifically for **membership privacy context**. The experimental results also showed that the proposed ALS achieve better privacy protection compared to the prior work.
>
> 2. **Regarding the experiment results.** We would like to clarify that, only HAMP is most comparable to the proposed ALS in terms of achieving a good privacy-balance trade-off. As detailed in the related work and Section 4.2, while HAMP offers the strongest balance between privacy protection and model utility, **ALS is significantly more efficient**. HAMP relies on high-entropy soft label generation and requires dataset-specific tuning of the entropy threshold to maintain accuracy. Furthermore, HAMP introduces additional computational overhead due to its test-time defense mechanism, which involves generating random samples and performing an additional inference step to determine the final output. We'd like to emphsize that **ALS not only achieve strong privacy-utility trade-off, but also is a simple and more efficient solution.**
>
> 3. **Regarding Proposition 3.1.** The purpose of the lower-bound proof in Proposition 3.1 is to establish the mathematical soundness and stability of the ALS loss, rather than to fully characterize its theoretical connection to generalization or privacy theory. As analyzed in Section 3.1, minimizing a standard cross-entropy loss drives the logits magnitude of training members arbitrarily large. In contrast, minimizing the ALS loss reduces membership distinguishability because both terms in the ALS objective (logit normalization and entropy regularization) explicitly prevent the model from producing extremely low loss or overly confident predictions on training (member) samples, which are the primary statistical signals exploited by membership inference.
>
>    1. **Logit normalization** $f/\|f\|$ and the temperature factor $1/\alpha$ bound the scale of the logits, so even correctly classified training samples cannot push the softmax probability arbitrarily close to 1.
>
>    2. The **entropy regularization** term $\lambda \mathcal{H}(\sigma(f/\|f\|))$ forces output distributions to maintain a minimum entropy level.
>
>    Together, they directly limit how low the loss can become on members reduces the confidence gap between members (low entropy) and non-members (higher entropy).
>
>    Proposition 3.1 shows that the lower bound of ALS objective. Our choices of $\lambda$ make sure it has a strict non-zero lower bound, meaning members can never reach the near-zero loss characteristic that uniquely identifies them. These mechanisms together mitigate the overconfidence issues for members and non-members, thereby reducing the statistical separability required for a successful membership inference attack.
>
>     Our focus, aligned with the current literature, is on providing comprehensive empirical evidence demonstrating that ALS consistently reduces membership-related distinguishability across diverse datasets and attacks. We agree that a broader theoretical framework that could analyze privacy leakage/defense is important. However, such analysis is beyond the scope of this work and we will consider it in future work.
>
> 4. **Regarding "standard" data samples.** We ackowledge and appreaciate the insights from [1]. However, our intention of evaluating the proposed ALS on the standard tabular and image data is primarily to ensure fair and controlled comparison with prior work[1,2,3], which also uses these datasets as standard benchmarks. In addition, our main focus and contribution of this work is to proposed a simple yet effective method to achieve better privacy-utility trade-off (high model uitility and low privacy leakage). However, we will consider evaluating more challenging data as suggested in the future work.

---

> > ### Author Response · Authors · 2025-11-24
> >
> > 5. **Regarding the reported utility degradation for RelaxLoss.** We would like to clarify that the choice of model architecture is different in the original paper of RelaxLoss and our work. Unlike existing baselines which consistently use Densenet, the original paper of RelaxLoss adopted ResNet-20 and VGG11for image data (CIFAR10 and CIFAR100) and MLP for tabular data (Purchase100 and Texas100). In contrast, we used Densenet 100 for image data following previous work. As reported in our paper, RelaxLoss can indeed maintain the model utility on the tabular dataset, while suffering utility degradation on image dataset. This potentially suggests that RelaxLoss does not generalize across model architectures. However, we additionally conducted further experiments using the experimental settings from RelaxLoss. Specifically, we adopt ResNet20 and train it on CIFAR10 and CIFAR100, respectively.
> >
> >    |  Datasets  |    CIFAR10    |  |   CIFAR100    |  |
> >    | :--------: | :-----------: | :----------------: | :-----------: | :----------------: |
> >    |  Defense   | Test Accuracy | Highest attack AUC | Test Accuracy | Highest attack AUC |
> >    | Undefended |     70.5      |       0.846        |     33.2      |       0.942        |
> >    | RelaxLoss  |     73.8      |       0.572        |     35.1      |       0.602        |
> >    | ALS(ours)  |     73.7      |       0.563        |     35.6      |       0.578        |
> >
> >    The table above presents the test accuracy and the highest attack AUC of Undefended, RelaxLoss, and our ALS, respectively. The results show that the model trained on ALS achieves 73.7 and 35.6 test accuracy on CIFAR10 and CIFAR100, comparable to or even better than RelaxLoss, which achieves test accuracy of 73.8 and 35.1, respectively. In addition, the hight attack AUCs of ALS are consistently lower than those of RelaxLoss, which suggests that ALS has better performance of preventing membership privacy leakage than RelaxLoss.
> >
> > We hope we have sufficiently addressed all of your questions, and are happy to engage further on more questions!
> >
> > [1] Shokri, Reza, et al. "Membership inference attacks against machine learning models." 2017.
> >
> > [2] X. Tang, S. Mahloujifar, L. Song, V. Shejwalkar, M. Nasr, A. Houmansadr, and P. Mittal, “Mitigating membership inference attacks by {Self-Distillation} through a novel ensemble architecture,” 2022
> >
> > [3] Chen, Zitao, and Karthik Pattabiraman. "Overconfidence is a dangerous thing: Mitigating membership inference attacks by enforcing less confident prediction.", 2023.

---

### Official Review · Reviewer_rhqr · 2025-11-01

**Soundness:** 2
**Presentation:** 2
**Contribution:** 1
**Rating:** 2
**Confidence:** 5

**Summary:**

The paper proposes Adaptive Logit Scaling, a modification to the standard Cross-Entropy loss to mitigate MIAs in machine learning models. ALS adaptively constrains the norm of output logits during training to reduce model overconfidence, thereby decreasing the distinguishability between member and non-member data. The authors validate their effectiveness through extensive experiments on four benchmark datasets. ALS is compared against eight state-of-the-art MIA defense methods, demonstrating superior performance in balancing privacy and model utility without significant computational overhead.

**Strengths:**

1. The solution is elegantly simple yet effective, requiring minimal changes to existing training pipelines.
2. The evaluation is thorough, testing across multiple datasets, attack types, and defense baselines.

**Weaknesses:**

1. It is recommended to include recent attack baselines, such as RMIA.
2. Fig.2 presents only a single privacy-utility point for each defense method, which is insufficient for a comprehensive evaluation of the privacy-utility trade-off. Additionally, one panel in Figure 2 combines different defense methods corresponding to various MIAs, which does not provide a justified assessment. I recommend plotting the privacy-utility curve in one panel for a certain attack, similar to RelaxLoss, to provide a more comprehensive evaluation of the trade-offs between privacy and utility.
3. The ablation studies in Table 3 are insufficient. I suggest including results from privacy-utility curves with varying values of $\alpha$ and $\lambda$ to better demonstrate the impact of these hyperparameters.
4. I question the claim that the model with a training accuracy of 97.06% (Purchase100 in Table 1) exhibits the entropy distribution centered at 4.605. If hyperparameters were tuned solely to achieve this result, it would undermine the significance, as most privacy defense methods can achieve similar outcomes. I recommend including the accuracy in Figure 1 and ensuring that the compared methods exhibit comparable test accuracy for a fair evaluation.

**Questions:**

Proposition 3.1 provides a lower bound for the loss. Could you elaborate on the theoretical connection between minimizing the ALS loss and the mechanism of reducing the distinguishability between member and non-member distributions?

---

> ### Author Response · Authors · 2025-12-03
>
> Dear reviewer,
>
> We are grateful for your valuable comments. We'd like to make the following clarifications.
>
> 1. **It is recommended to include recent attack baselines, such as RMIA.**
>
>    We additionally evaluate our method using RMIA. Following their implementation, we adopt 2^7 reference models in the offline setting. The results are shown in the table below.
>
>    |    Metric    | Undefended | MemGuard | DPSGD | LS   | HAMP | RelaxLoss | LogitNorm | ALS(ours) |
>    | :----------: | :--------: | :------: | :---: | ---- | ---- | :-------: | :-------: | :-------: |
>    | TPR@0.01%FPR |    3.67    |   1.96   | 0.32  | 2.54 | 0.63 |   0.59    |   0.68    |   0.41    |
>    | TNR@0.01%FNR |   12.23    |   8.21   | 0.48  | 6.12 | 0.52 |   0.81    |   0.74    |   0.58    |
>
>    Consistent with the main results, these findings further demonstrate that ALS provides stronger defense performance under RMIA. While DPSGD achieves low leakage, it does so at the cost of severely degraded model utility. Compared with HAMP, RelaxLoss, and LogitNorm, ALS achieves lower leakage and thus offers a more effective privacy defense.
>
> 2. **Regarding the privacy-utility trade-off and different attacks for each defense method in Figure 2. ** We would like to clarify that the privacy-utility trade-off we refer to in the paper highlights the challenges of achieving strong privacy protection without significantly compromising the model utility, which we have clearly mentioned in the paper. Figure 2 exactly presents this property for each defense method. In addition, as stated and justified clearly in the caption of Figure 2, **“since we report the best-performing attack per defense, the selected attacks may differ.”** This choice avoids underestimating any defense method by ensuring that each method is evaluated under its strongest adversary. We also provided detailed results of each attacks on all four datasets in Table 5 and Table 6 in the Appendix D. Overall, ALS consistently provides both **strong privacy protection** for members and non-members and **high model utility**, achieving a more favorable privacy–utility trade-off compared with existing defenses—even when each is evaluated against its strongest attack.
>
> 3. **Regarding the ablation study on the hyperparameter.** We would like to note that the effects of the hyperparameters **λ** and **s** have already been thoroughly examined in **Table 2** and **Figure 5**, which analyze sensitivity and performance as these parameters vary. These results already demonstrate how the privacy/utility outcomes change across different settings.
>
> 4. **Regarding the entropy distribution in Figure 1 and model accuracy. ** The entropy distributions shown in Figure 1 are directly computed from the trained models used in our experiments. As reported in Table 1, the undefended model, LogitNorm, and ALS all achieve comparable test accuracies (i.e., 80.85%, 81.95%, 81.1% of test accuracy on Purchase100, respectively),  which ensures a fair comparison of their entropy distributions as suggested.
>
> 5. **Regarding the theoretical connection between minimizing ALS and MIA privacy protection.** MIAs exploit overconfidence and distributional skew between member samples and non-member samples. As analyzed in Section 3.1, minimizing a standard cross-entropy loss drives the logits magnitude of training members arbitrarily large. In contrast, minimizing the ALS loss reduces membership distinguishability because both terms in the ALS objective (logit normalization and entropy regularization) explicitly prevent the model from producing extremely low loss or overly confident predictions on training (member) samples, which are the primary statistical signals exploited by membership inference.
>
>    1. **Logit normalization** $f/\|f\|$ and the temperature factor $1/\alpha$ bound the scale of the logits, so even correctly classified training samples cannot push the softmax probability arbitrarily close to 1.
>
>    2. The **entropy regularization** term $\lambda \mathcal{H}(\sigma(f/\|f\|))$ forces output distributions to maintain a minimum entropy level.
>
>    Together, they directly limit how low the loss can become on members abd therefore reduces the confidence gap between members (low entropy) and non-members (higher entropy).
>
>    Proposition 3.1 shows the lower bound of the ALS objective. Our choices of $\lambda$ make sure it has a strict non-zero lower bound, meaning members can never reach the near-zero loss characteristic that uniquely identifies them. These mechanisms together mitigate the overconfidence issues for members and non-members, thereby reducing the statistical separability required for a successful membership inference attack.
>
> We hope we have sufficiently addressed all of your questions, and are happy to engage further on more questions!

---

### Official Review · Reviewer_YhN7 · 2025-11-01

**Soundness:** 3
**Presentation:** 3
**Contribution:** 3
**Rating:** 2
**Confidence:** 4

**Summary:**

This paper proposes a novel defense mechanism called Adaptive Logit Scaling (ALS) loss function. It aims to mitigate the overconfidence of deep learning models by constraining the norm of output logits, thereby reducing privacy leakage in Membership Inference Attacks. Unlike traditional cross-entropy loss functions, ALS prevents the model from making overconfident predictions on training samples by dynamically adjusting the scale of logits for each sample during training and introducing an entropy regularization term to encourage high-entropy outputs. This reduces the distinguishability between member and non-member data. Experimental results show that ALS, evaluated against eight MIA attacks and nine baseline defense methods, demonstrates superior privacy protection performance while maintaining high model accuracy.

**Strengths:**

Strongness
1、Innovation and Practicality: The ALS loss is a simple yet effective defense mechanism. It can be implemented by merely modifying the loss function, requiring no additional data or complex operations during the inference phase. It significantly enhances privacy protection capabilities without causing substantial degradation in model performance. This lightweight design enables easy integration into existing training pipelines, meeting the needs of practical applications.
2、Comprehensiveness of Experimental Design: The paper evaluates eight MIA attacks (including state-of-the-art LiRA and label-only attacks) and compares ALS with nine other defense methods across four datasets, covering both privacy metrics and model utility. Consistent results show that ALS reduces privacy leakage while the drop in accuracy is negligible, demonstrating its robustness.
3、Theoretical Support and Cross-Domain Value: The paper derives the theoretical lower bound of the ALS loss function, providing mathematical proof for the rationality of the method. Although further deepening is possible, this theoretical work already surpasses most empirical defense studies.

**Weaknesses:**

Weaknesses
1、Insufficient Depth in Comparison with Existing Work: Although the paper compares ALS with nine defense methods, it does not thoroughly discuss the core differences between ALS and similar approaches (e.g., LogitNorm). For instance, LogitNorm uses fixed norm scaling, while ALS emphasizes adaptiveness. However, the paper only provides qualitative demonstration through Figure 1, without quantitative analysis of the advantages of this adaptiveness on specific samples.
2、Limitations in Experimental Scope: The paper mainly focuses on image and tabular datasets, failing to cover extreme data scenarios such as small-sample datasets, highly imbalanced category data, or high-noise data. It also does not involve more sensitive domains (e.g., medical or text data) nor test large-scale models (e.g., Transformers). Additionally, despite the comprehensive attack evaluation, adaptive attack scenarios (e.g., attackers being aware of the defense mechanism) are not considered, which may overestimate the actual protection capability.
3、Limited Depth in Theoretical Analysis: While the paper provides a proof for the lower bound of the ALS loss, it does not deeply explore its connection with generalization ability or privacy theory. For example, the lower bound only depends on the number of categories k and the hyperparameter λ, with no analysis of its robustness under complex models or distribution shifts. This may limit the theoretical universality of the method.

**Questions:**

n.a

---

> ### Author Response · Authors · 2025-11-24
>
> Dear reviewer,
>
> We are thankful for your comments. We would like to make the following clarifications:
>
> 1. **Regarding the comparison with existing work.** We would like to clarify that Figure 1 is based on quantitative statistics, not qualitative visualization. Specifically, we compute the entropy (or maximum confidence) for every member/non-member sample and then plot their empirical distributions. The figure shows that LogitNorm still produces larger non-overlapping regions between member and non-member distributions, while ALS yields significantly more overlap, indicating reduced distinguishability. In addition, the experimental results consistently show that ALS achieves stronger privacy protection than LogitNorm across all benchmark attacks, which confirms the benefit of the adaptiveness introduced by ALS.
>
>    However, to further compare our method against LogitNorm, we analyze the hardest and easiest samples based on entropy. Specifically, we select the top 10% highest-entropy samples (hard samples) and the bottom 10% lowest-entropy samples for both LogitNorm and ALS. We then compute the average entropy for member and non-member samples within these subsets.
>
>    For the top 10% highest-entropy samples:
>
>    |  Defense  |  Member  | Non-member | $\Delta$ |
>    | :-------: | :------: | :--------: | :------: |
>    | LogitNorm | 4.583317 |  4.590638  | 0.007321 |
>    |    ALS    | 4.605093 |  4.605101  | 0.000008 |
>
>    For the bottom 10% lowest-entropy samples:
>
>    |  Defense  |  Member  | Non-member | $\Delta$ |
>    | :-------: | :------: | :--------: | :------: |
>    | LogitNorm | 4.297603 |  4.319160  | 0.021557 |
>    |    ALS    | 4.604581 |  4.604610  | 0.000029 |
>
>    The results from the tables above show that ALS consistently produces almost lower entropy gap for member and non-member samples, with $\Delta$ values close to zero. In contrast, LogitNorm exhibits a higher entropy gap especially among the easiest samples, suggesting that its output still retains distinguishable patterns between member and non-member data. Together with the attack evaluation results presented in the main paper, these findings suggest that ALS provides stronger privacy protection, effectively reducing the distinguishability between member and non-member samples.
>
> 2. **Regarding the experimental scope and data diversity.** We focus on image and tabular datasets primarily to ensure fair and controlled comparison with prior work, which also uses these datasets as standard benchmarks[1,2,3]. Our choice of attack methods likewise follows previous literature to maintain consistency and comparability. We acknowledge that evaluating small-sample, highly imbalanced, or high-noise datasets, as well as extending to text or medical domains, is an important future direction. However, these scenarios are orthogonal to the core contribution of the paper and can be explored in follow-up work without affecting the validity of the proposed method.
>
> 3. **Regarding the depth of theoretical analysis.** The purpose of the lower-bound proof in Proposition 3.1 is to establish the mathematical soundness and stability of the ALS loss, rather than to fully characterize its theoretical connection to generalization or privacy theory. We agree that a broader theoretical framework would strengthen the universality of the method. However, such analysis is non-trivial and beyond the scope of this work. Our focus, aligned with the current literature, is on providing comprehensive empirical evidence demonstrating that ALS consistently reduces membership-related distinguishability across diverse datasets and attacks.
>
> We hope we have sufficiently addressed all of your questions, and are happy to engage further on more questions!
>
> [1] Shokri, Reza, et al. "Membership inference attacks against machine learning models." *2017 IEEE symposium on security and privacy (SP)*. IEEE, 2017.
>
> [2] X. Tang, S. Mahloujifar, L. Song, V. Shejwalkar, M. Nasr, A. Houmansadr, and P. Mittal, “Mitigating membership inference attacks by {Self-Distillation} through a novel ensemble architecture,” in 31st USENIX Security Symposium (USENIX Security 22), 2022, pp. 1433–1450
>
> [3] Chen, Zitao, and Karthik Pattabiraman. "Overconfidence is a dangerous thing: Mitigating membership inference attacks by enforcing less confident prediction." *arXiv preprint arXiv:2307.01610* (2023).

---

### Meta-Review · Area_Chair_sKNC · 2026-01-13

**Summary:**

Reviewers liked the simplicity of the method but they all had concerns about the novelty of the work. The reviewers also pointed that the experiments need to be more through with better comparison with recent attacks and defenses.

**Reviewer Concerns:**

Authors provide new experiments in their rebuttal. This addresses some of the concerns on comparison with previous defenses and provides new ablations. The concerns about technical novelty and not considering adaptive attacks are still outstanding.

**Reviewer Scores:**

Reviewer YhN7 may raise their score given the new quantitative comparison with logitNorm.
Reviewer 3gnU may raise their score given the new ablation experiments.

---

### Decision · Program_Chairs · 2026-01-26

Reject